# Frail Older Adults without Occupational Dysfunction Maintain Good Subjective Well-Being: A Cross-Sectional Study

**DOI:** 10.3390/healthcare10101922

**Published:** 2022-09-30

**Authors:** Keisuke Fujii, Yuya Fujii, Yuta Kubo, Korin Tateoka, Jue Liu, Koki Nagata, Daiki Nakashima, Tomohiro Okura

**Affiliations:** 1Department of Occupational Therapy, Faculty of Health Sciences, Kansai University of Health Sciences, 2-11-1 Wakaba, Kumatori, Osaka 590-0482, Japan; 2Physical Fitness Research Institute, Meiji Yasuda Life Foundation of Health and Welfare, 150 Tobuki, Tokyo 192-0001, Japan; 3Division of Occupational Therapy, Faculty of Rehabilitation and Care, Seijoh University, 2-172 Fukinodai, Tokai 476-8588, Japan; 4Doctoral Program in Physical Education, Health and Sport Sciences, Degree Programs in Comprehensive Human Sciences, Graduate School of Comprehensive Human Sciences, University of Tsukuba, 1-1-1 Tennodai, Tsukuba 305-8577, Japan; 5Doctoral Program in Public Health, Degree Programs in Comprehensive Human Sciences, Graduate School of Comprehensive Human Sciences, University of Tsukuba, 1-1-1 Tennodai, Tsukuba 305-8577, Japan; 6Department of Rehabilitation, Faculty of Health Science, Naragakuen University, 3-15-1 Nakatomigaoka, Nara 631-8524, Japan; 7Faculty of Health and Sport Sciences, University of Tsukuba, 1-1-1 Tennodai, Tsukuba 305-8577, Japan; 8R&D Center for Tailor-Made QOL, University of Tsukuba, 1-1-1 Tennodai, Tsukuba 305-8577, Japan

**Keywords:** occupational therapy, occupational dysfunction, well-being, quality of life, frail, frailty

## Abstract

The purpose of this cross-sectional study was to clarify the relationship between frailty/occupational dysfunction (OD), both with and without, and subjective well-being among community-dwelling older adults. A total of 2308 (average age: 72.2 ± 5.1, female: 47.0%) independently living older adults in Kasama City, Japan, completed a self-administered questionnaire in November 2019. OD, frailty, and subjective well-being were assessed. Participants were classified into six groups: robust and healthy occupational function (HOF), robust and OD, pre-frail and HOF, pre-frail and OD, frail and HOF, and frail and OD. To examine the relationship between frailty/OD and subjective well-being, we performed an analysis of variance with subjective well-being as the dependent variable and with and without frailty/OD (six groups) as the independent variables. The results showed a significant association between with and without frailty/OD and subjective well-being in community-dwelling older adults. The subjective well-being of the pre-frail and HOF group was significantly better than that of the robust and OD group. Furthermore, the subjective well-being of the frail and HOF group was significantly better than that of the pre-frail and OD group. These results can be used to develop a new support method for frailty.

## 1. Introduction

The increasing aging of the population is a serious problem worldwide, and frailty in older age is an important issue. Frailty is associated with meaningful life events in old age, such as long-term care needs [1], hospitalization [1], and mortality [1,2,3]. Frailty is classified into three categories: “robust,” “pre-frail,” and “frail” [4]. In Japan, frail and pre-frail rates are 7.5–9.9% and 38.7–42.9%, respectively, and the total frail and pre-frail rates reach 50% [5]. Therefore, appropriate measures are essential to address frailty in older adults.

The direction of traditional measures against frailty has been to improve or prevent frailty, and approaches to improvement have mainly been from “pre-frail to robust” or “frail to pre-frail” [6,7,8,9,10,11]. However, in some cases, age-related functional decline is unavoidable in old age. Therefore, it is necessary to consider the health support for frailty from a different perspective. A key outcome of health support in old age is the quality of life and well-being. Therefore, it is necessary to improve the quality of life and well-being of frail older adults, not only to improve frailty but also to improve how they lead a satisfying life even if they are frail.

Occupational therapists are professionals who deal with difficulties in daily living; that is, occupational dysfunction (OD) [12]. Previous studies have reported an association between OD and quality of life [12]. Therefore, those without OD may have higher quality of life and well-being than those with OD. In addition, OD is a modifiable factor that could be improved even in hospitalized or disabled patients who have more functional decline than frail patients [13,14]. This means that even those who are in a state of frailty owing to inevitable functional decline caused by aging and other factors may be able to improve their quality of life and well-being by improving their OD. Further, frail older adults without OD may have a higher quality of life and well-being than those with OD.

This study examined the relationship between those with and without frailty/OD and well-being among community-dwelling older adults. The findings inform the development of health support from the perspective of OD for those who have difficulty in improving frailty in the preventive occupational therapy field.

## 2. Materials and Methods

### 2.1. Participants and Data Collection

This cross-sectional study used data from the Kasama Study 2019, which was conducted in Kasama, Ibaraki, Japan, in collaboration with the city of Kasama in 2013, 2014, and 2017 to investigate preventive care strategies. As of 1 October 2019, the population of Kasama City was 74,334 with an aging rate of 31.7% [15]. Kasama City is located in the Kanto area of Japan. It lies 100 km northeast of Tokyo, Japan’s capital city.

Figure 1 shows the participant flowchart. The inclusion criteria for this study were (1) those aged 65–85 years and (2) those who did not require nursing care. A total of 8000 participants were randomly selected from the basic resident register on 1 October 2019. The self-administered questionnaires were mailed by the researchers and Kasama City office staff in November 2019. Responses were received from 3934 persons (response rate: 49.2%). The exclusion criteria were those who (1) were hospitalized at the time of the survey; (2) had a history of stroke, dementia, or psychiatric disorder; and (3) had missing data. All participants were informed of the study details in writing, and their voluntary return of the completed questionnaire was considered consent to participate in the study. This study was approved by the ethics committee of the University of Tsukuba (no. Tai 019–101).

### 2.2. Measurement Variables

#### 2.2.1. Definition of Frailty

Frailty was evaluated using the Kihon Checklist (KCL) [16]. The KCL consists of 25 questions about instrumental activities of daily living (IADL; 3 questions), social ADL (4 questions), physical functions (5 questions), nutritional status (2 questions), oral function (3 questions), cognitive function (3 questions), and mood status (5 questions), which were answered with “yes” or “no” responses (0–25 points). Higher KCL scores indicated frailty: 0–3 points = normal, 4–7 points = pre-frail, and 8 points or more = frail. The validity of the KCL frail criteria has been verified to be sufficiently related to the frailty criteria by the Cardiovascular Health Study criteria [17].

#### 2.2.2. Occupational Dysfunction

OD was assessed using the Classification and Assessment of OD (CAOD). It comprises 16 questions related to difficulties in daily living [12], with responses rated on a seven-point Likert scale from “strongly agree” (7) to “strongly disagree” (1). The total CAOD score ranges from to 16–112, with higher scores indicating more severe OD. A cutoff score of 52 or higher was defined as OD.

#### 2.2.3. Subjective Well-Being

Subjective well-being has been used to assess happiness [18,19,20]. Based on previous studies, subjective happiness was measured using the following question: “In general, how happy or unhappy do you usually feel?” with 11 response options ranging from 0 (“very unhappy”) to 10 (“very happy”).

#### 2.2.4. Demographic Data

Demographic data were collected and used as covariates, including sex, age, education history, subjective economic status, IADL, and social interaction status. Subjective economic status was assessed using the question, “How do you feel about your current economic situation?” Responses were rated on a range of “very difficult,” “slightly difficult,” “normal,” “somewhat rich,” and “very rich.” The two categories of “very difficult” and “slightly difficult” were operationally defined as “poor.” IADL was evaluated using the five items of the Tokyo Metropolitan Institute of Gerontology Index of Competence (TMIG-IC) based on subjective evaluations by respondents [21]. The TMIG-IC is a commonly used assessment in Japan and can measure higher-level functional capacity in community-dwelling older adults. The five items of IADL are scored on a five-point scale, with higher scores indicating better IADL. In this study, having a TMIG-IC total score of less than 5 points was defined as IADL disability [22,23]. Social interaction status was evaluated using the Japanese version of the Lubben Social Network Scale short version (LSNS) [24]. The LSNS consists of six items: three related to the number of people in the family network and three related to the number of people in the friends and acquaintances network. Responses were rated on a six-point scale (range: 0–30 points). A lower total score corresponded to poorer social interaction.

### 2.3. Statistical Analysis

According to the cutoff value of CAOD and KCL, participants were divided into the “robust and healthy occupational function (HOF) group (CAOD score ≤ 51 points and KCL score 0–3 points),” “robust and OD group (CAOD score ≥ 52 points and KCL score 0–3 points),” “pre-frail and HOF group (CAOD score ≤ 51 points and KCL score 4–7 points),” “pre-frail and OD group (CAOD score ≥ 52 points and KCL score 4–7 points),” “frail and HOF group (CAOD score ≤ 51 points and KCL score ≥ 8 points),” and “frail and OD group (CAOD score ≥ 52 points and KCL score ≥ 8 points).”

Means and standard deviations were calculated for continuous variables and frequencies and percentages for categorical variables. To examine the relationship between those with and without frailty/OD and subjective well-being, we performed an analysis of variance with subjective well-being as the dependent variable and with and without frailty/OD (six groups) as the independent variable. Two models were used in this study, a crude model and an adjusted model. The latter was adjusted for age, sex, educational history, subjective economic status, IADL ability, and social interaction status. These covariates have been suggested to be potential confounders in previous studies. When significant differences were found, the Bonferroni method was used for multiple comparisons. Statistical analysis was performed using IBM SPSS Statistics 28.0 (IBM, Armonk, NY, USA). In this study, a *p*-value < 0.05 was considered significant.

## 3. Results

Participants’ characteristics are presented in Table 1. The percentage of robustness was 57.9% (*n* = 1336), 30.7% (*n* = 709) were pre-frail, and 11.4% (*n* = 263) were frail. The proportion of participants with OD was 14.2% (*n* = 327). The subjective well-being of the robust participants was 7.5 ± 1.8 points, 6.9 ± 1.9 points for the pre-frail, and 6.2 ± 2.1 points for the frail. Robust individuals with OD accounted for 7.6% (*n* = 101), pre-frail for 19.5% (*n* = 138), and frail for 33.5% (*n* = 88).

Table 2 shows the association between those with and without frailty/OD and subjective well-being. In the crude model, a significant association was found between with and without frailty/OD and subjective well-being. Multiple comparison tests showed that the robust and HOF group had significantly better subjective well-being than all other groups. The pre-frail and HOF group had significantly better subjective well-being than the robust and OD, pre-frail and OD, frail and HOF, and frail and OD groups. The frail and HOF group had significantly better subjective well-being than the pre-frail and OD and frail and OD groups. The robust and OD group had significantly better subjective well-being than the frail and OD group. Further, the adjusted model showed a significant association between with and without frailty/OD and subjective well-being (Figure 2). Multiple comparison tests showed that the robust and HOF group had significantly better subjective well-being than the other groups, with the exception of the pre-frail and HOF group. The pre-frail and HOF group had significantly better subjective well-being than the robust and OD, pre-frail and OD, and frail and OD groups. The frail and HOF group had significantly better subjective well-being than the pre-frail and OD and frail and OD groups.

## 4. Discussion

This cross-sectional study showed a significant association between those with and without frailty/OD and subjective well-being among community-dwelling older adults. After adjusting for potential covariates, the frail and HOF group had higher subjective well-being than the frail and OD and pre-frail and OD groups, and the pre-frail and HOF group had higher subjective well-being than the pre-frail and OD and robust and OD groups. Thus, our results suggest that frailty, but not OD, preserves higher subjective well-being.

First, subjective well-being was higher for robust, pre-frail, and frail individuals. This result is similar to that of previous studies that examined the relationship between physical health and well-being [25]. The proportion of participants with OD was higher for frail, pre-frail, and robust individuals, respectively. To the best of our knowledge, this is the first report of an association between frailty and OD. Our findings suggest that approximately 8% of robust older adults may have OD. Furthermore, approximately 20% of pre-frail and 35% of frail older adults have OD, suggesting that the assessment of OD is important in old age, especially among the frail.

Regardless of the adjustment for covariates, the results of multiple comparison tests showed that the subjective well-being of the HOF groups was significantly higher than that of the OD groups at each frailty stage. Previous studies have confirmed that OD is associated with quality of life [12]. In this study, similar results were obtained for subjective well-being, which is an outcome measure similar to quality of life. The results revealed that the association between OD and subjective well-being did not change at any stage of frailty, which is a novel finding. The absence of OD was associated with higher subjective well-being at each frailty stage. Furthermore, the subjective well-being of the pre-frail and HOF group was significantly higher than that of the robust and OD group, and the subjective well-being of the frail and HOF group was significantly higher than that of the pre-frail and OD group. This finding is the greatest strength of this study. In previous studies, the percentage of improvement from frail to pre-frail and from pre-frail to robust in observational studies was 38.8% [26], and in intervention studies aimed at improvement, the improvement rate was reported to be 31.4% [10]. Therefore, the improvement rate of frailty in observational and intervention studies was about 40%, and more than half of the patients did not show any change. However, well-being is an important indicator for successful aging. Therefore, it is necessary to maintain higher well-being, even when it is difficult to improve frailty. It may be necessary to focus on OD to maintain high well-being. OD, in addition to being associated with quality of life, such as frailty, is a modifiable factor. In this study, the presence of OD was more important than frailty, as the pre-frail and HOF group retained higher subjective well-being than the robust and OD group. Although the reason for this result is unclear, it is possible that the impact of OD on subjective well-being was stronger than that of frailty. Further research is needed to clarify the impact of OD on subjective well-being.

This study had several limitations. First, as this was a cross-sectional study, it was not possible to confirm causal relationships. Therefore, a longitudinal study is necessary. Second, this study did not use the standard methods of assessing quality of life and well-being, such as the SF-36. However, subjective well-being, which is validated and easy to use [20], was considered to be useful in this study. Third, although the participants were randomly selected, the number of responses was less than 50%, and sampling bias may have been present. Fourth, this study was conducted in Japan. It is expected that people’s lives are influenced by culture, and the same is true for occupational dysfunction. Therefore, it is difficult to generalize the findings of this study to other countries.

Despite these limitations, the findings indicate that even for those who are frail or pre-frail and have difficulty improving frailty, an approach to reversible OD may contribute to maintaining high subjective well-being. Longitudinal and intervention studies are needed to examine the impact of frailty and OD on subjective well-being.

## 5. Conclusions

This study found a significant association between with and without frailty/OD and subjective well-being in community-dwelling older adults. In addition, pre-frail and frail older adults who did not have OD maintained good subjective well-being. Strategies need to be developed to not only improve frailty but also reduce OD to maintain good well-being among older adults. However, these findings do not mean giving up on improving frailty. Future interventions for occupational dysfunction are expected while continuing interventions for improving/preventing frailty. To improve occupational dysfunction, it is necessary to modify an individual’s lifestyle. Therefore, it is desirable to develop a program that combines the perspectives of both frailty and occupational dysfunction, such as an exercise and nutrition intervention that is effective in improving/preventing frailty, followed by a health lecture to reconsider one’s own lifestyle and improve occupational dysfunction.

## Figures and Tables

**Figure 1 healthcare-10-01922-f001:**
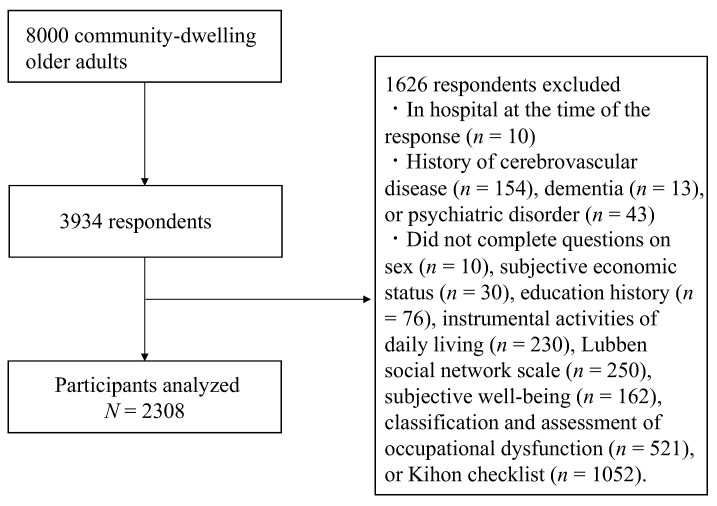
Participant flowchart.

**Figure 2 healthcare-10-01922-f002:**
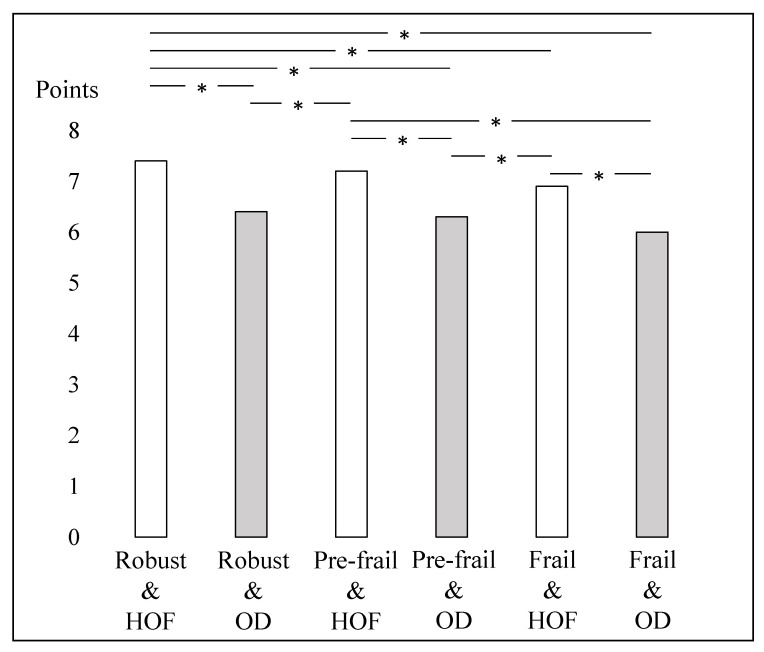
With and without frailty/OD and subjective well-being (adjusted model). * *p* < 0.05. HOF: healthy occupational function, OD: occupational dysfunction.

**Table 1 healthcare-10-01922-t001:** Participants’ characteristics.

Frailty	Robust	Robust	Pre-Frail	Pre-Frail	Frail	Frail
Occupational dysfunction	HOF	OD	HOF	OD	HOF	OD
*n*	1235	101	571	138	175	88
Age (years), mean ± SD	71.5 ± 4.7	71.5 ± 4.9	72.8 ± 5.0	71.8 ± 5.3	75.1 ± 5.8	73.9 ± 6.0
Female, % (*n*)	48.2 (595)	45.5 (46)	46.8 (267)	51.4 (71)	38.9 (68)	43.2 (38)
Education history (≥ high school), % (*n*)	88.5 (1093)	85.1 (86)	81.8 (467)	83.3 (115)	77.7 (136)	87.5 (77)
Subjective economic status (poor), % (*n*)	12.0 (148)	24.8 (25)	22.1 (126)	25.4 (35)	34.3 (60)	44.3 (39)
IADL ability (disability), % (*n*)	4.0 (50)	4.0 (4)	10.3 (59)	5.1 (7)	23.4 (41)	14.8 (13)
LSNS (points), mean ± SD	17.9 ± 5.6	15.7 ± 6.0	15.7 ± 5.7	14.3 ± 5.4	12.9 ± 5.6	12.5 ± 6.2
Subjective well-being (points), mean ± SD	7.5 ± 1.7	6.3 ± 1.8	7.1 ± 1.9	6.1 ± 1.9	6.5 ± 2.0	5.5 ± 2.0
CAOD (points), mean ± SD	28.5 ± 10.3	58.9 ± 6.2	31.4 ± 10.4	59.8 ± 7.3	35.9 ± 10.0	63.6 ± 10.4
Kihon checklist (points), mean ± SD	1.5 ± 1.1	1.8 ± 1.0	5.0 ± 1.0	5.4 ± 1.1	9.8 ± 2.2	10.6 ± 2.7

HOF: healthy occupational function, OD: occupational dysfunction, SD: standard deviation, IADL: instrumental activities of daily living, LSNS: Lubben social network scale, CAOD: classification and assessment of occupational dysfunction.

**Table 2 healthcare-10-01922-t002:** With and without frailty/OD and subjective well-being.

	Crude Model			Adjusted Model		
	Mean (95% CI)	ANOVA *p*	Post hoc Test with Bonferroni Correction	Mean (95% CI)	ANCOVA *p*	Post hoc Test with Bonferroni Correction
Robust and healthy occupational function (A)	7.5 (7.4–7.6)	<0.001	A > B, C, D, E, F C > B, D, E, F E > D, F B > F	7.4 (7.3–7.5)	<0.001	A > B, D, E, F C > B, D, F E > D, F
Robust and OD (B)	6.3 (6.0–6.7)			6.4 (6.1–6.8)		
Pre-frail and healthy occupational function (C)	7.1 (7.0–7.3)			7.2 (7.1–7.4)		
Pre-frail and OD (D)	6.1 (5.8–6.4)			6.3 (6.0–6.6)		
Frail and healthy occupational function (E)	6.5 (6.3–6.8)			6.9 (6.6–7.2)		
Frail and OD (F)	5.5 (5.1–5.8)			6.0 (5.6–6.3)		

CI: confidence interval, OD: occupational dysfunction. The adjusted model was adjusted for age, sex, education history, subjective economic status, instrumental activities of daily living, and the Lubben social network scale.

## Data Availability

The data presented in this study are available upon request from the corresponding author. The data were not publicly available because of privacy or ethical restrictions.

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
