# Peer review of "Frail Older Adults without Occupational Dysfunction Maintain Good Subjective Well-Being: A Cross-Sectional Study"

_healthcare, 2022, doi:10.3390/healthcare10101922_

Round 1
Reviewer 1 Report
The paper entitled ‘Frail older adults without occupational dysfunction maintain good subjective well-being: A cross-sectional study” discusses frailty/occupational dysfunction, an issue of particular importance to the elderly people.
As a reviewer I would like to ask the authors to revise the following:
1. It would be beneficial to include in the Abstract section the aim of the study and research methods that were used.
2. It would be advisable to put the tables and figures in the Results section in a more orderly fashion.
Author Response
Thank you very much for reviewing our paper. We attach a response letter.

Reviewer 2 Report
I really enjoyed reading this manuscript, the case is well presented and concise.
I have very few comments/suggestions:
Line 47, 'pre-failure' should this be 'pre-frail'?
Lines 77-78 - Please consider adding the geographic location of Kasama City, (which region?) within Japan, so the reader can get a sense of where this city is located.
Conclusion - Please consider elaborating on the strategies you deem are needed to both improve frailty and reduce OD to maintain well-being amongst older people. Also, authors could elaborate on the implications of the study results on health and social care practice for community-dwelling older adults.
Author Response

(The authors gave the same response as above.)

Reviewer 3 Report
Authors should clarify the recruitment of the sample as well as how the questionnaires were sent. who provided the emails and who sent them?
when did the participants give their consent to receive the questionnaire?
figure 1 should be moved to chapter
Author Response

(The authors gave the same response as above.)
